# Skin-interfacing wearable biosensors for smart health monitoring of infants and neonates
Lauren Zhou [1,2], Matthew Guess [1,2], Ka Ram Kim [1,2] & Woon-Hong Yeo [1,2,3,4] ✉

Health monitoring of infant patients in intensive care can be especially strenuous for both the patient and their caregiver, as testing setups involve a tangle of electrodes, probes, and catheters that keep the patient bedridden. This has typically involved expensive and imposing machines, to track physiological metrics such as heart rate, respiration rate, temperature, blood oxygen saturation, blood pressure, and ion concentrations. However, in the past couple of decades, research advancements have propelled a world of soft, wearable, and non-invasive systems to supersede current practices. This paper summarizes the latest advancements in neonatal wearable systems and the different approaches to each branch of physiological monitoring, with an emphasis on smart skin-interfaced wearables. Weaknesses and shortfalls are also addressed, with some guidelines provided to help drive the further research needed.

The advancements in the miniaturization of electronics have allowed impressive achievements toward improving healthcare. Wearable sensing technology enables comfortable, continuous, and convenient alternatives to standard care while maintaining quality and accuracy. These systems are especially valuable for neonatal applications, where small footprints, gentle handling, and ease of use are essential. Traditional intensive care within the Neonatal Intensive Care Unit (NICU) involves a complicated mess of electrodes, tubing, and tape that is visually disturbing and makes it extremely difficult for kangaroo care, which is skin-to-skin swaddling reported to have numerous physiological, behavioral, and therapeutic benefits[1]. In addition, neonatal skin is thin and fragile, making it especially susceptible to irritation, stripping, and sores, which may result in permanent scarring and damage[2]. With improvements in miniaturization, material choices, fabrication methods, signal analysis techniques, and wireless communication, new wearable sensors and systems that alleviate many of the inconveniences of conventional care have been developed (Fig. 1). In this review, we will detail the material basis for wearable devices. Then, we will share the underlying principles and technological progress of essential branches of physiological monitoring, including biopotential, optical, temperature, electrochemical, and multi-signal sensing. Finally, we will identify deficiencies that should be addressed for future work to improve the equity and accessibility of quality healthcare.

## Material developments
### Design considerations for neonates

There are several nuances when designing for infant patient groups. First, infants are undeniably smaller than adults, with most high-risk patients suffering from prematurity (gestational age <37 weeks) and low birth weight (typically <2.5 kg)[3]. These patients undergo several monitoring options, including cardiac, respiratory, neurological, and general physiological monitoring[4]. Blood draws for diagnostic tests are also standard, subjecting neonates to an average of ~7.5–17.3 painful procedures per day[5]. These monitoring practices require expensive specialty equipment, with electrodes and tubing needing to be secured with adhesive tapes that are applied and removed multiple times daily. This can be especially damaging and dangerous to infant groups, as the stratum corneum and epidermis layers are 30% and 20% thinner than adult skin[6], and the cohesion between the dermis and epidermis is much weaker[2,6]. This reduced skin function results in a high risk of epidermal stripping, contact dermatitis, pressure wounds, tension blisters, and burns (Fig. 2a, b)[2,7], and leads to a greater risk of infections due to percutaneous invasion of pathogens[8]. Thus, when it comes to modernizing this technology and making it wearable, the critical design considerations include small and conformal form factors, gentle and biocompatible adhesion to the skin, and making the design appear benign to caregivers, all while maintaining excellent signal quality and effective operation.

[1]George W. Woodruff School of Mechanical Engineering, Georgia Institute of Technology, Atlanta, GA 30332, USA. [2]IEN Center for Wearable Intelligent Systems and Healthcare, Institute for Electronics and Nanotechnology, Georgia Institute of Technology, Atlanta, GA 30332, USA. [3]Wallace H. Coulter Department of Biomedical Engineering, Georgia Institute of Technology and Emory University School of Medicine, Atlanta, GA 30332, USA. [4]Parker H. Petit Institute for Bioengineering and Biosciences, Institute for Robotics and Intelligent Machines, Georgia Institute of Technology, Atlanta, GA 30332, USA.
✉e-mail: whyeo@gatech.edu

**Fig. 1 | Illustration of the working principles towards translating conventional monitoring practices to wearable form factors.** Methods include electronic miniaturization, soft and flexible materials, gentler adhesion mechanisms, designing all-in-one devices, prioritizing non-invasive monitoring and increased accessibility, wireless communication via Bluetooth, NFC, or radio, and cloud-based data processing. Stock image of infant outline adapted with permission from purchase by Getty Images iStock with color modifications and device adaptations added by the authors.

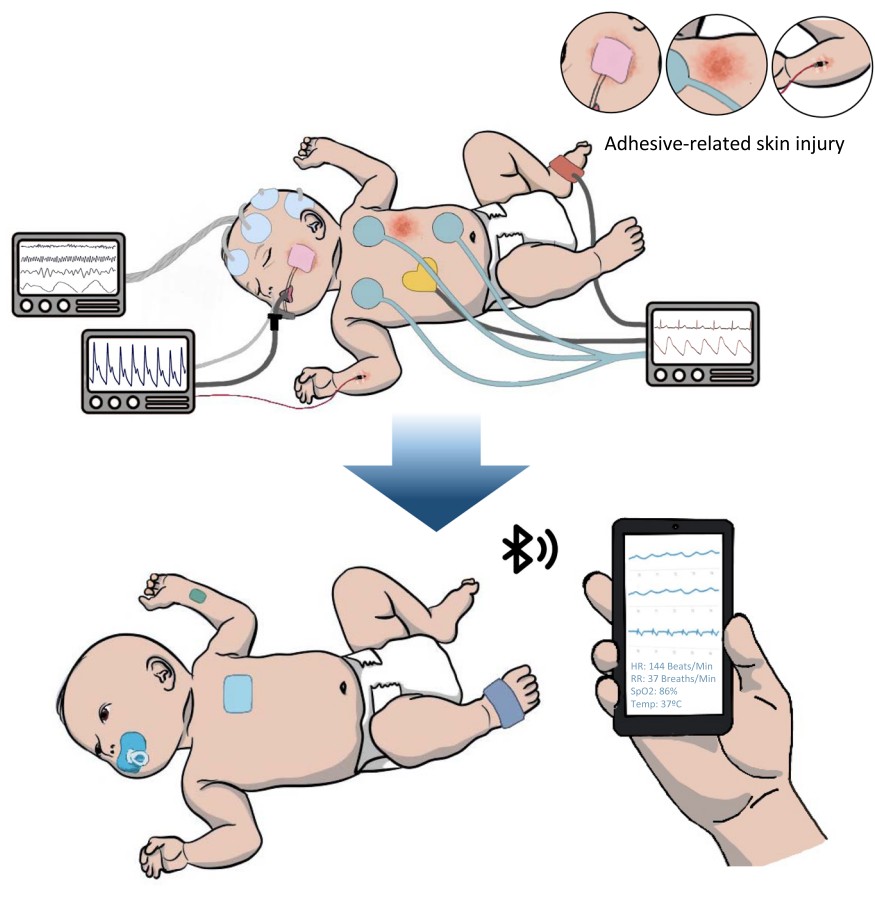

Adhesive-related skin injury

## Epidermal electronic systems

One of the primary technological techniques to combat these issues is called "epidermal electronic systems" (EES), where the material properties of the system match that of the epidermis[9]. The skin surface topology is imperfect, with creases, pores, and general surface texture forming gaps between the electrode-skin interface, resulting in increased impedance, motion artifacts, and worsened signal acquisition. EES systems are soft and flexible electronics that are fabricated using microelectromechanical systems (MEMS) fabrication techniques; however, while typical MEMS methods with silicon wafers produce delicate and brittle devices[10], EES approaches allow robust flexibility. Employing thin film materials like polyimide (PI) (<100 μm thick) as a structural substrate and dielectric material and thin depositions of metals like Cr/Al/Cu/Au (<500 nm thick) to serve as the conductive layers[9,11,12] one can achieve ultrathin and flexible devices with sub-nanometer bending stiffness and effective moduli ~140 kPa[9], preventing delamination from the skin. They can be easily attached to the skin with a thin adhesive transfer film like that of a temporary tattoo[9] or via liquid bandage (Fig. 2c)[11] with very intimate contact with the skin's surface (Fig. 2d)[11]. Integrating modified geometric designs using curvilinear serpentine interconnects enables greater strain tolerance and stretchability as they act as pre-buckled lines. High amplitude serpentines are capable of 100% strain with minimal stress and maintain elastic responses (Fig. 2e)[13], surpassing the elastic behavior of skin where it is only linearly elastic up to tensile strains of 15%[14]. The high conformality allows the system to move dynamically with the skin, assures prominent signal quality, eliminates motion artifacts, and maintains continuous contact with the skin. The thin-film properties of these electronics allow comfortable and gentle physiological monitoring that protects the integrity of fragile infant skin while ensuring high-quality data recordings.

Thin film systems can have broader applications when interfaced with elastomers. More affordable than silicon wafer technology, they can have varying Young's modulus depending on the amount of crosslinking allowed, and they are unique in that they can form tight seals with itself, silicon, and glass. Tight seals make the devices water resistant, which is necessary in high humidity NICU incubators, and they can be injected with ionic liquid to serve as a decoupling layer to reduce mechanical stresses within the device (Fig. 2f)[12]. This quality also makes elastomers excellent for micro and nanofluidic applications for biofluid sensing which will be discussed later in this review. For applications with EES, elastomers are an excellent substrate to embed soft electronics within because they are biocompatible and naturally adhesive due to Van der Waals forces. Typical adhesive tapes used in the NICU to secure tubing and wires are pressure-based and have tackifiers derived from acrylate, resin, or petroleum. After applying pressure to the tape onto the skin, the adhesive flows into the creases and folds of the skin via wetting and form a bond, building strength over time[15,16]. Because the adhesive forms a strong bond with the skin, there is a very high likelihood that it is stronger than the bond between the skin cells, causing the epidermal layers to be stripped away with removal. This makes the elastomeric Van der Waals adhesion especially attractive because it can bond strongly to the skin (even wet) while allowing extremely gentle removal (Fig. 2g). Compared to other standard adhesives used in the NICU designed to be delicate on the skin, Ecoflex has a peel force 15 and 10 times smaller than Tegaderm and Kind Removal tape, respectively (Fig. 2h)[17]. In a clinical test on 50 neonates observing changes in skin condition (erythema, dryness, or breakdown) after 15 minutes of EES application, the average change was negligible to slightly improved[17]. Thicker layers of elastomer prevent EES disturbance (Fig. 2i) and depending on the elastomer type, can be reusable, improving device longevity. Alternative adhesives that emphasize and improve on gentle removal processes are also being explored, like trigger-detachable hydrogels that swell and lose their adhesive energy once treated with glucose[18] (Fig. 2j), water[19], or shear force[20], or thermally switchable copolymer tapes[21] and silicone-based adhesive with

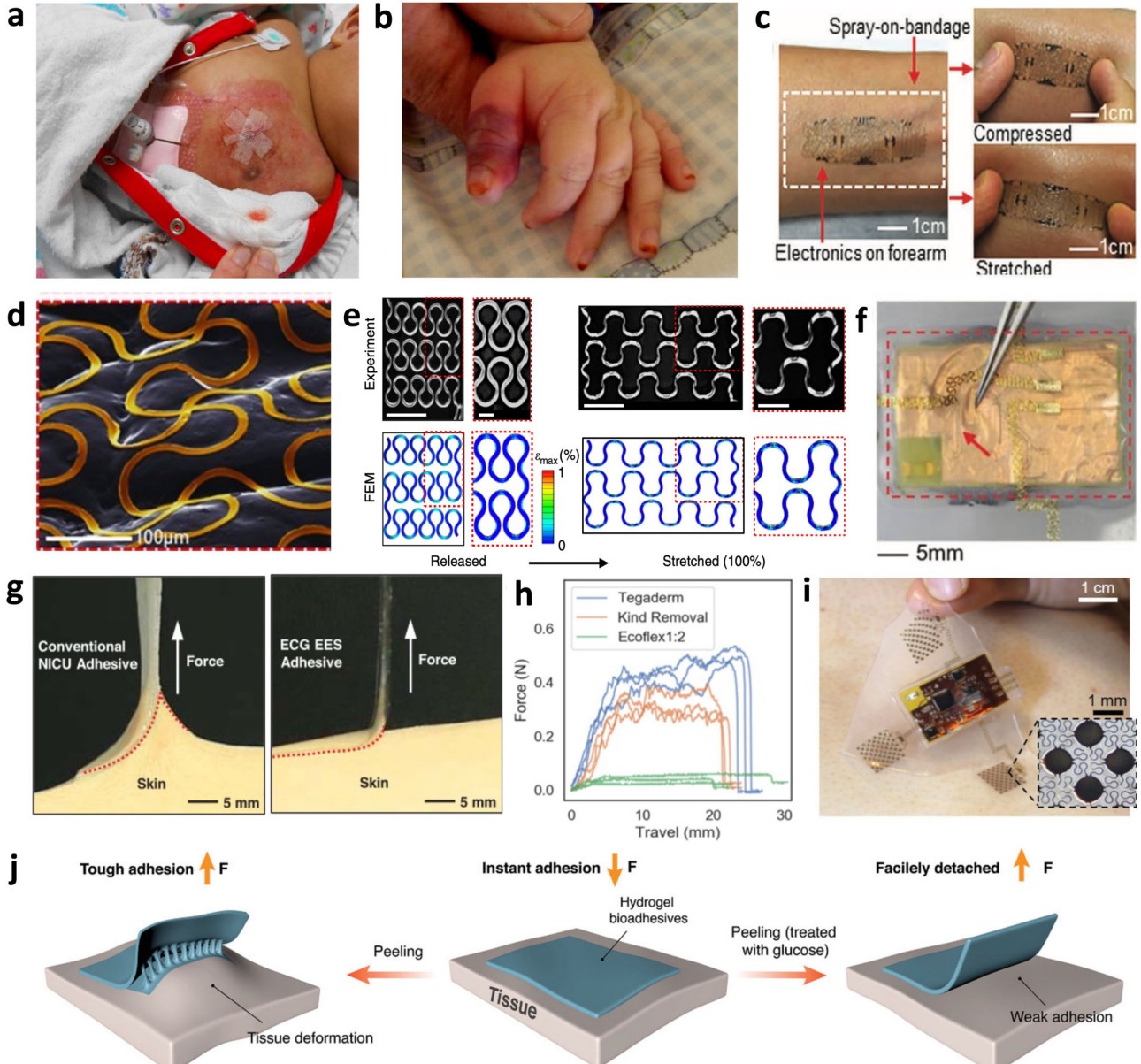

**Fig. 2 | Materials and design choices for wearable electronics. a** Contact dermatitis from repeated adhesive wet electrode placement. Reproduced with permission[2]. Copyright 2014, Elsevier Inc. **b** Second-degree burn from pulse oximeter clip. Reproduced with permission[94]. Copyright 2012, Elsevier Inc. **c** Robust and conformal adhesion of EES via liquid bandage. Adapted with permission[11]. Copyright 2013, WILEY-VCH Verlag GmbH. **d** High EES conformality to skin texture. Reproduced with permission[11]. Copyright 2013, WILEY-VCH Verlag GmbH. **e** Geometric serpentine lines allowing linearly elastic response up to 100% strain. Reproduced with permission[13]. Copyright 2014, Springer Nature Limited. **f** Ionic liquid-filled decoupling layer within EES device. Reproduced under the terms of the Creative Commons CC BY license from ref. 12. **g** Comparison of peel force between conventional NICU adhesive and elastomeric EES adhesive. Reproduced under the terms of the Creative Commons CC BY 4.0 license from ref.33. **h** Peel force comparison between Ecoflex and common NICU tapes Tegaderm and Kind Removal tape. Reproduced with permission[32]. Copyright 2020, IEEE. **i** Image demonstrating thicker elastomeric layers prevent irreparable EES deformation. Reproduced under the terms of the Creative Commons CC BY 4.0 license from ref. 12. **j** Diagram illustrating glucose trigger-activated hydrogel adhesive that loses adhesion after wetting. Reproduced under the terms of the Creative Commons CC BY 4.0 license from ref. 18.

meltable oil crystallites[22]. These alternative and gentler removal processes may be better suited for infants with especially fragile skin, like premature infants.

## All-in-one systems

Depending on the criticality of a patient, they may be connected to several monitors simultaneously, each with its own apparatus. These complicated setups form a perceived barrier of untouchability, curtailing skin-to-skin contact and exacerbating visual disturbance and stress. Elastomeric encapsulation is excellent for hybrid electronics, allowing the mix of flexible EES sensors, rigid passive and active electronics, wireless communication, and data processing to form all-in-one devices with real-time wireless monitoring. These self-sufficient devices are the basis for several systems described in this review.

## Textile sensors

Textiles are a popular mechanism by which to integrate sensors for biosignal sensing. From a review of the literature for noninvasive infant monitoring, textile sensors were historically the preferred mechanism due to its perceived familiarity and simplicity. Integration methods of

conductive materials include direct weaving of electrically conductive fibers[23,24], sewing on pieces of Ag- or Au-coated nylon[25] or polyurethane[26], or printing electric inks on the fibers directly[27]. These components can then be easily attached to items of clothing like onesies[24], straps/bands[28], jackets[29], and more. Textile electrodes embedded within clothes eliminates the need for tapes and conductive gels, but at the cost of signal quality. Physiological metrics like respiration rate and motion can be measured by the resistive and capacitive strain response of conductive fibers[24,28]. Most of these works transfer data with wired connectors or bulky wireless transmitters; however, antennas can be knitted within clothing coupled with radio frequency identification (RFID) tags for battery-free wireless data transfer[30]. We found that as EES systems and flexible electronics were developed and advanced for adult monitoring, they had become the state of the art for infant monitoring as well; thus, it will be the main point of focus for this review.

## Physiological monitoring

The measurement of physiological metrics can be performed over various modalities, including biopotential, optical, temperature, electrochemical, and multi-signal sensing. Here, we report gold standard of testing and non-invasive skin-interfaced alternatives applicable to neonatal patient groups, summarized in Table 1.

### Biopotential sensing

The human body functions via chemical reactions that generate action potentials to power and control the physiological processes within the body. Several of these biopotentials are measurable through the skin with electrodes that transduce the ionic current into assessable electric signals. Several biopotential signals that are significant for health monitoring including electrocardiograms (ECG) for heart activity, electromyograms for muscle activation, electroencephalograms (EEG) for brain activity, and electro-oculograms for eye movement tracking, with ECG and EEG being most relevant for neonatal applications. Traditional wet Ag/AgCl electrodes use conductive gels (sometimes abrasive) to reduce the impedance made by the nonconformal gaps between the electrode and skin surface and improve electrical conductivity. However, these gels dry out over time degrading signal quality, irritating the skin, and still requiring the use of a strong adhesive to affix the electrode. Furthermore, conventional electrodes are wired to stationary monitors that restrict movement. Therefore, recent works to improve biopotential sensing have focused on using EES as dry electrodes that do not require the use of conductive gels, which mitigate and improve on these weaknesses.

**Electrocardiography.** Electrocardiography (ECG), which measures the electrical activity of the heart muscles, is a key signal used to monitor heart function by being able to derive heart rate (HR), heart rate variability (HRV), heart rhythms, and respiration rate (RR). To determine if the signal recorded is high quality, ECG fiducials and timings including the P, QRS, and T waves should be distinct and consistent without distortion. To minimize device footprints, lead placement is modified to place the electrodes closer to one another. With the adapted lead placement, it is important to ensure maintained accuracy. To extract real-time heart rate from the ECG signal, the Pan-Tompkins algorithm is a widely used method to identify the QRS complex which is then used by an automated algorithm to quantify HR, HRV, heart rhythm, and RR (Fig. 3a)[31,32]. By recursively checking if the heart and respiratory rates are within normal range, emergency alarms can sound to alert the provider of abnormal behavior, thus meeting the standards of NICU monitoring. Chung et al.[33] were the first to apply an all-in-one EES system to mimic vital signs monitoring in the NICU (Fig. 3b). Semiconductor fabrication steps form the circuital base of the device, including the electrodes, near field communication (NFC) coil, and sensor interconnects. Off-the-shelf active and passive components for biopotential amplification and filtering and an NFC System-on-a-Chip (SoC) allow wireless inductive power transfer and data sharing to a host reader platform that lays beneath the patient's mattress. An ionic fluid injected into a microfluidic space beneath the circuitry maintains the resonant frequency and quality factor for the NFC antenna coil. In addition, the battery-free, open mesh design allows device usage during medical imaging (MRI and X-ray). Although only two electrodes are used, they had strong agreement in their HR and RR determination to the gold standard; however, they still required the use of a conductive gel. Weaknesses of this system is that it is mechanically fragile, collapsing on itself after extreme deformation during the removal process[9,17]. Due to it complicated fabrication steps requiring specialized facilities, these expensive systems are not suitable for popular and disposable use. In addition, the NFC communication protocol only has modest operating distances up to 25 cm, keeping the infant bedridden for continuous monitoring. Kim et al.[12] optimized the construction strategy for EES to preserve high conformality while improving the system hardiness, finding that a mixed elastomer with a 2:1 ratio of EcoFlex Gel to EcoFlex 0030 had the greatest adhesion force and conformability with robust manipulability. They applied this discovery to their own neonatal ECG monitoring device (Fig. 3c)[32], which had a removable lithium-ion polymer (LiPo) battery allowing hours of continuous monitoring and a Bluetooth low energy (BLE) SoC allowing long-range telemetry up to 15 meters. Using a modified Lead V2 configuration, they were able to easily distinguish the ECG fiducial PQRST waves and had an SNR over 40 dB without the use of conductive gels. This work also used semiconductor fabrication steps including spin coating, sputter deposition, chemical vapor deposition, wet/dry etching, and photolithography to form the device. Textile electrodes are also popular for ECG recording (Fig. 3d) without conductive gels and have relatively good performance with low drift and distinguishable QRS complexes[29]. However, constant pressure needs to be applied, either by having the infant laying on the electrodes or with a tight-fitting jacket. With wireless communication and smart computation with machine learning, these device systems can perform real-time HR, RR, HRV, and heart rhythm determination. However, the real-time study of heart rhythms from wirelessly recorded systems should be taken fastidiously, as issues with inconsistent data transfer like data drop out and lagging can give artificial results.

**Electroencephalography.** Electroencephalography (EEG) measures the electrical activity of the brain with electrodes placed on the scalp, typically used for seizure detection. Common modes are single-channel amplitude integrated EEG (aEEG) and multichannel continuous EEG (cEEG), where cEEG is considered the diagnostic gold standard. aEEG can easily be performed on premature infants to observe abnormalities associated with brain injury[34] while cEEG, requiring ~10-20 electrodes, is more comprehensive and used for seizure detection and cortical function assessment[35]. Due to incubator limitation, head size constraints, and hair, placement and maintenance of the electrodes is difficult. In addition, EEG waveforms are very small in amplitude and susceptible to artifacts. A wireless communication device helps allow space-saving communication between the 23-lead system and a bedside laptop[36], however limited research has been performed to materially redesign an EEG electrode system. Most developments involve designing textile caps or bands that improve the ease of electrode placement but still use traditional wet electrodes[37–39]. Although not designed for neonatal applications, Mullen et al.[40] designed a rigid cEEG headset using novel hybrid electrodes that have an ionic hydrogel sandwiched between a semi-permeable membrane and Ag/AgCl plates to transduce the electrical signal. This hybrid electrode design combines the improved conductivity and signal quality of wet electrodes with the skin biocompatibility of dry electrodes. Bristle electrodes, shaped like the bristles of a brush with a conductive tip, may also be a suitable dry electrode replacement, as they can part the hair for gentle conformal contact with the skin and record with high signal quality[41]. This design, however, has been reported as irritating after long term use as a constant pressure is required for low skin-electrode impedance.

**Table 1 | Comparison chart of non-invasive wearable equivalents to conventional methods for popular NICU tests**

| Metric/Test | Conventional Method | Wearable Alternative | Performance Compared to Conventional Device | Reference |
|---|---|---|---|---|
| Heart rate, heart rate variability, heart rhythm | Wired gel electrodes | Wireless ECG | High correlation | 15,17,32,33 |
| | | Wireless pulse oximeter | High correlation, but sensitive to motion artifacts | 47,48 |
| Respiration | Wired gel electrodes | Wireless ECG | High correlation | 15,17,32,33 |
| | | Wireless pulse oximeter | High correlation, but sensitive to motion artifacts | 44 |
| Blood oxygen saturation, hemoglobin | Wired probe to pulse oximeter | Wireless pulse oximeter | High correlation | 15,33,47,48 |
| Cerebral hemodynamics | Cerebral ultrasound, functional MRI (fMRI), CT scan | Wireless NIR | High correlation with fMRI | 58 |
| Core Temperature | Invasive arterial line, Rectal/esophageal probe | ZHF sensor | Moderately high correlation to esophageal probe | 66 |
| | | Transistor | Needs further validation | 70 |
| Blood Pressure | Invasive arterial line | Wireless Multi-signal system | High correlation but frequent calibration is required | 15,33 |
| Cerebral Palsy Detection | MRI, Wired EEG | Wireless Multi-signal system | Moderate accuracy, further clinical validation is necessary | 58,81,82 |
| Jaundice detection | Blood draw/urine test | Wireless optical colorimetry | High correlation | 59 |
| | | Wireless noninvasive electrochemical diaper (urine) | Not clinically validated | 77 |
| Cystic fibrosis detection (Chloride monitoring) | Blood draw | Wireless noninvasive electrochemical sensor patch (sweat) | High correlation | 75,93 |
| Phenylketonuria Detection | Blood draw | Wireless noninvasive electrochemical wristband (saliva) | High correlation, further clinical validation is necessary | 74 |
| Glucose monitoring | Blood draw/urine test | Wireless noninvasive electrochemical sensor patch (sweat) | High correlation | 75,76 |
| Saliva sampling | Swab | Wireless noninvasive electrochemical pacifier (saliva) | High correlation | 72 |
| Sodium monitoring | Blood draw | Wireless noninvasive electrochemical pacifier (saliva) | High correlation | 73 |
| Potassium monitoring | Blood draw | Wireless noninvasive electrochemical pacifier (saliva) | High correlation | 73 |

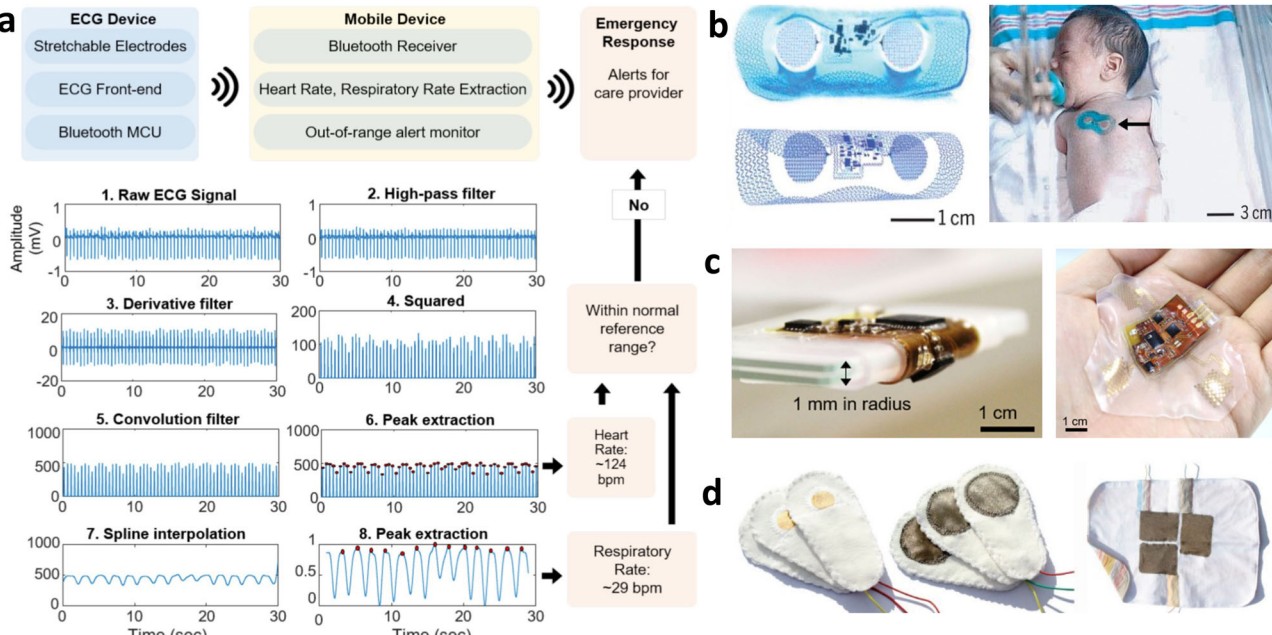

**Fig. 3 | Mechanisms and devices for biopotential sensing. a** Process diagram for heart rate and respiration rate determination from ECG waveform following Pan-Tompkins algorithm. The calculated values can be studied recursively for alert systems. Reproduced with permission[32]. Copyright 2020, IEEE. **b** First neonatal all-in-one EES system (left) for neonatal cardiovascular monitoring (right). Reproduced under the terms of the Creative Commons CC BY 4.0 license from[33]. **c** Thicker, more robust EES cardiovascular monitoring with V2 lead setup. Reproduced with permission[32]. Copyright 2020, IEEE. **d** Textile-based electrodes. Reproduced under the terms of the Creative Commons CC BY 4.0 license from ref. 29.

## Optical sensing

Optical sensing for medical use uses the principles of spectrophotometry to noninvasively observe physiological phenomena transcutaneously. Using pairs of light emitting diodes (LEDs) with corresponding photodetectors, the LED shines a wavelength with a known intensity into the tissue that then gets received by the photodetector. Based on how much light gets absorbed, one can form a representative relationship for key tissue metrics like blood oxygenation, hydration, and chemical composition[42]. There are two modes that exist: transmissive (LED and photodetector on opposite sides of the observed tissue) and reflective (LED and photodetector are side-by-side).

**Photoplethysmography.** Photoplethysmography (PPG) is an optical sensing technique used to estimate blood oxygen saturation (SpO2), also called pulse oximetry (POX). The gold standard of POX is with an invasive arterial line to find arterial oxygen saturation (SaO2), but optically derived values from POX are acceptable as well. POX is a dominant metric used for clinical decision making, including the decision to supply additional oxygen, the detection of early sepsis and cardiopulmonary complications, and is the sole screening metric for congenital heart defects (CHDs)[43]. POX can track respiration rate and indicate apnea events in premature infants, as cessations of breathing are accompanied by either bradycardia or oxygen desaturation (SpO2 < 80%)[44]. Using two sets of LED/photodetector pairs, one LED operates at a 640–660 nm wavelength (red) while the other at 880–940 nm (infrared)[45] to measure the differential light absorption through the observed skin tissue. This ratio is calibrated against SaO2 measurements to establish a measurement of SpO2. Typical POX probes for adults are designed for the finger and utilize the transmissive mode, often in a clip-style packaging. However, the strong forces of the clip can be especially damaging on fragile infant anatomy (Fig. 2b). Thus, neonatal pulse oximeters operate with the reflective mode on the hand or foot and are often packaged within an adhesive wrap. POX is highly susceptible to motion artifacts, and because infants frequently swing around their limbs, packaging for these sensors are designed to induce a constant force and must be checked regularly to avoid pressure wounds and burns. In addition, due to their

small anatomy and low body fat, neonates often have low blood perfusion in their limbs, which is necessary for accurate measurement. In these cases, clinicians will monitor from more proximal locations like the earlobe and forehead[46]. Grubb et al.[47] proposed forehead reflectance PPG for neonates in the NICU. Henry et al.[48] presented a wireless, cap-mounted device for measuring heart rate via PPG. The device was optimized for preterm infants by miniaturizing and ruggedizing the sensor, which consisted of four 525 nm LEDs arranged in pairs on either side of a photodetector. The sensor was encapsulated in silicone and wired to a computer for data recording. The sensor was held to the forehead using a specialized T-shaped cap (Fig. 4a). They demonstrated the clinical capabilities of the device by comparing the heart rate from ECG. Chung et al.[33] developed a flexible PPG sensor for the foot that utilized ultra-thin electronics to allow a small bending radius of 5 mm to allow for conformality with the foot (Fig. 4b). The thinness and design of the perforations of the device allowed for adhesion via only van der Waals forces. The PPG device offered on-board AC/DC calculation before transmitting data, reducing the bandwidth needed. The group built on this work in 2020 by introducing a system that improved on the limitations of the previous system (Fig. 4c)[17]. This system improved the wireless data communication operating distance, the fragile nature of the designs, and specialized fabrication. The PPG unit was made with a flexible printed circuit board encapsulated with silicone, providing more stability. Due to the embedded battery, the full PPG waveform could be transmitted over the device with 95% limits of agreement (LOA) in HR of less than 4 BPM with standard clinical measurement. There exist global weaknesses with POX for infants that need to be addressed. First, skin color is a big factor, as individuals with darker skin absorb more light than lighter skin[49]. In an assessment among adult patients of different races and ethnicities, Black, Hispanic, and Asian patients experienced erroneous over-estimations by POX compared to SaO2 levels[50]. This is likely due to the historical limited inclusion of people of color in these formative studies, where calibration algorithms are derived from data sets primarily from individuals with lighter skin tones[51]. Additional research should also be conducted for modifying the calibration algorithms for infant patients, as a study found

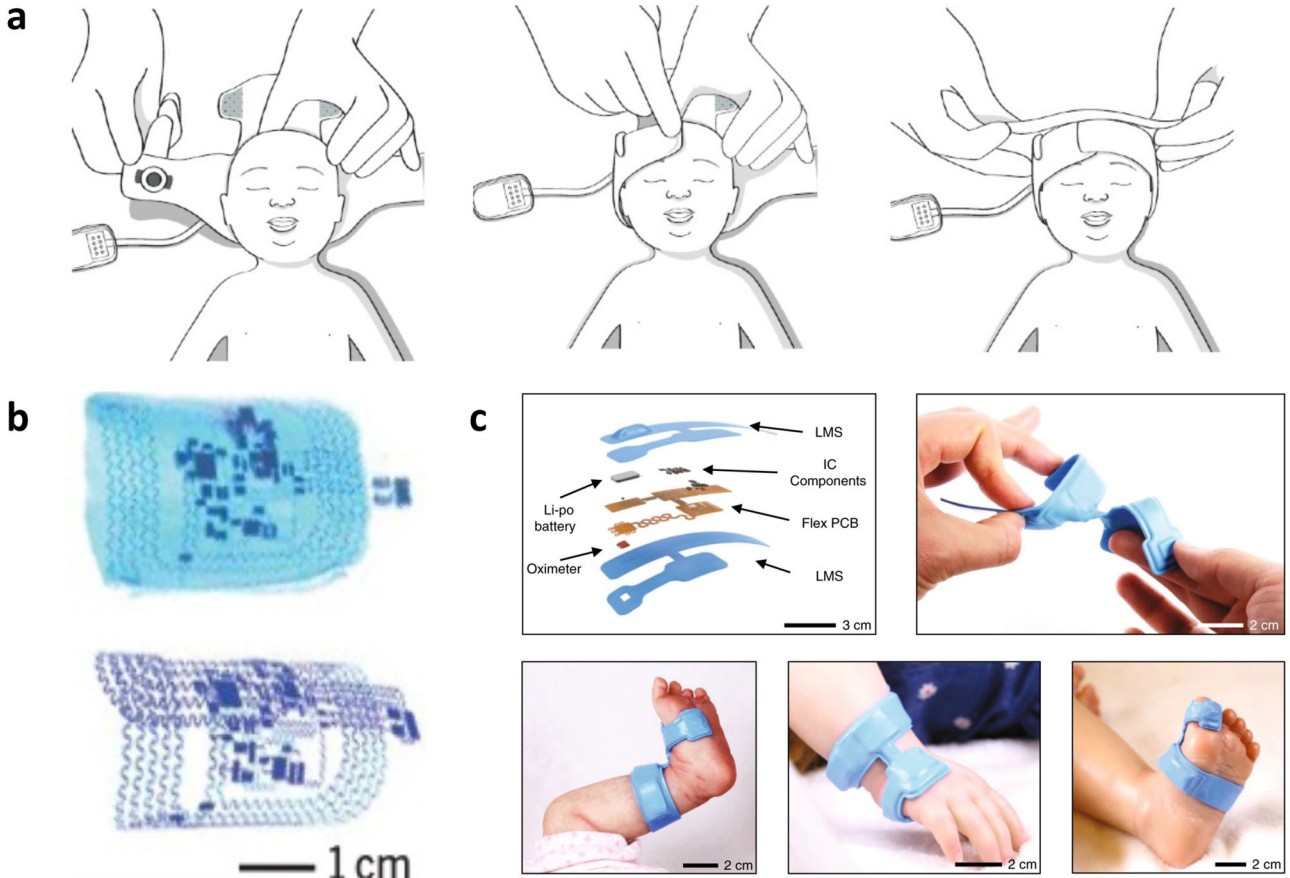

**Fig. 4 | Devices for optical sensing. a** Illustration of a cap-mounted device for measuring PPG at the forehead. The sensor sits above the eyebrow. Reproduced under the terms of the Creative Commons CC BY 4.0 license from ref. 48. **b** Wireless PPG device bending around a glass cylinder. Reproduced under the terms of the Creative Commons CC BY 4.0 license from ref. 33. **c** Schematic and photographs of a wireless limb unit for measuring PPG. Reproduced with permission[17]. Copyright 2020, Springer Nature.

that conventional NICU pulse oximeters lose accuracy for hypoxic patients with blood saturations lower than 85%, often overestimating by more than 5% compared to arterial values[52]. Diseased neonatal patients that suffer from CHDs often have blood oxygen levels in the 75–85% region, therefore these overestimations are unacceptable for regular clinical care. Given POX's clinical influence, these issues must be addressed with future research developments.

**Near-infrared spectroscopy.** Complementary to POX is near-infrared spectroscopy (NIRS). NIRS is based on the near-infrared spectrum of wavelengths (700–1000 nm) and the absorption of chromophores such as myoglobin, hemoglobin, and cytochrome aa3[53]. Although similar in principle to POX, NIRS differs in that it represents the balance of local tissue oxygen supply and demand. Regional tissue oxygen saturation (rSO2) monitors discriminate light paths from different tissue depths, ergo measuring veinous as well as arterial hemoglobin oxygenation[54]. Therefore, NIRS can be more useful in measuring cerebral hemodynamics, which is helpful for seizure detection[55], intraventricular hemorrhage[56], and white matter injury[57]. Rwei et al. developed a soft, flexible, wireless system for monitoring infant cerebral hemodynamics[58]. The pair of LEDs emit at 740 nm and 850 nm wavelengths with four photodiodes at source-detector distances of 5, 10, 15, and 20 mm, allowing recording at different tissue depths for detection of both peripheral and cerebral hemodynamics, and was validated by Monte Carlo optical simulations and magnetic resonance imaging (MRI). Other wavelengths offer opportunities to measure different chromophores. If blue and green wavelengths of LEDs are used, the light absorption ratio of bilirubin can be monitored, and this principle was used by Inamori et al.[59]

who reported a wearable device for jaundice detection. Calibrated by a commercial bilirubinometer, their device could successfully measure bilirubin concentration using the ratio of the reflected green and blue lights. Measuring NIRS offers challenges not found in photoplethysmography. The validation of NIRS is difficult because there is not a gold standard for tissue oxyhemoglobin content. Light absorption and scattering occurs from compounds other than hemoglobin, thus more wavelengths and source-detector distances are necessary for improved accuracy[60].

**Temperature sensing**

Infants, due to their prematurity and lack of body fat, struggle to maintain a stable and normal body temperature between 36.5 and 37.5 °C[61], thus are at extremely high risk of hypothermia. Premature infants are at even greater risks, which result in incubators stays tuned to the ideal temperature and levels of oxygen, humidity, and light. Skin and core temperatures are the two metrics studied. For skin temperature, single-measurement readings can be determined from a thermometer or human touch. For continuous readings, small flexible adhesive probes attached to the abdomen, chest, or back are used. Most wearable systems that have been developed for infants record skin temperature with off-the-shelf temperature gauges[17,19,62]. However, skin temperature is not very informative as a health metric because it cannot be used as a proxy for core temperature[63,64]. For core body temperature (CBT), common methods include measuring invasively from the pulmonary artery or semi-invasively with probes inserted into the esophagus, nasopharynx, tympanic membrane, or rectum[65]. However, these measurements still vary amongst each other depending on anatomical location because some organs produce heat (brain, liver) while others dissipate (lungs, skin)[66].

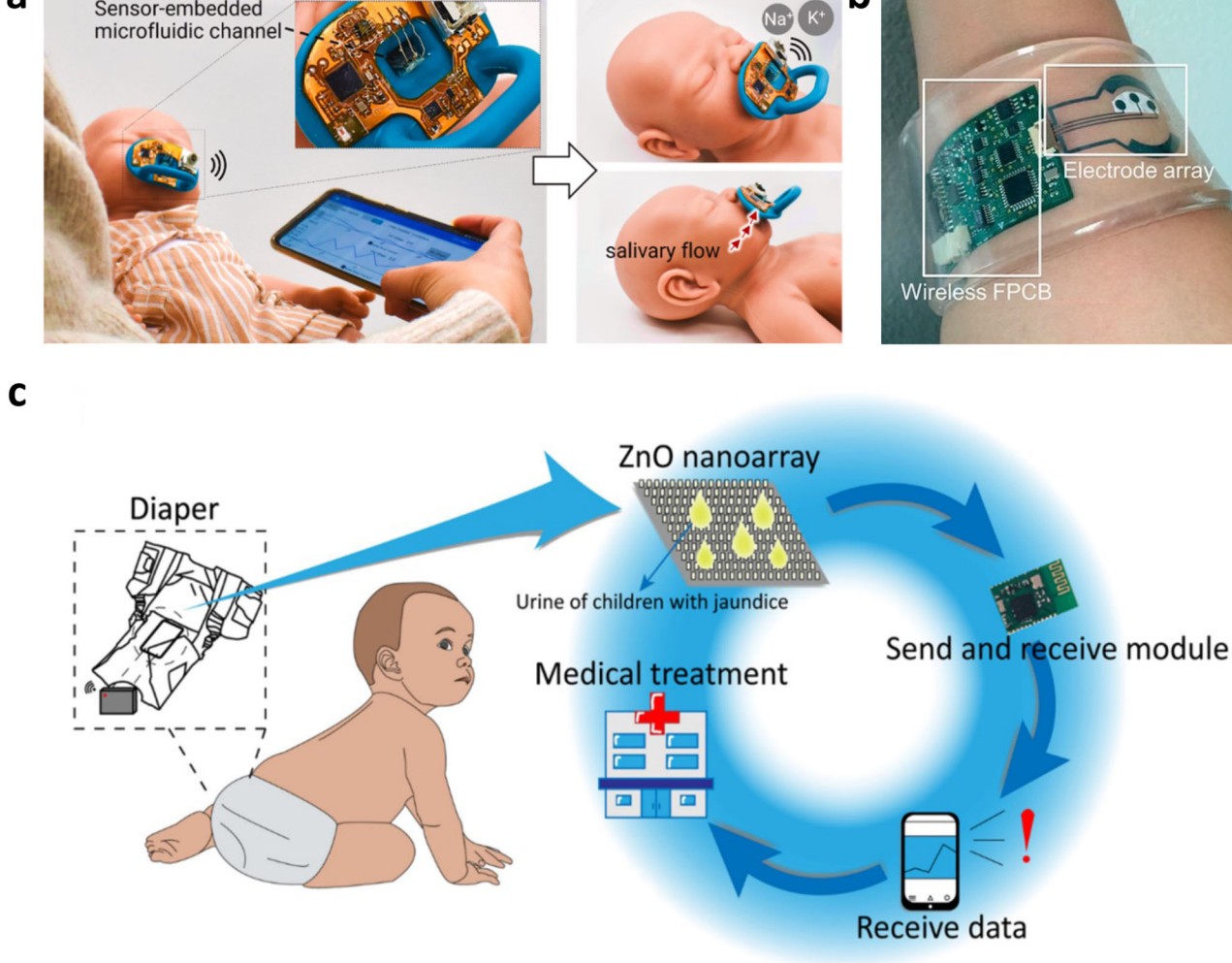

**Fig. 5 | Devices for electrochemical sensing. a** Electrochemical sensing pacifier that tracks electrolyte (sodium and potassium) levels from saliva. Reproduced with permission[73]. Copyright 2022, Elsevier B.V. All rights reserved. **b** Patch-style sweat sensor for sodium and chloride monitoring for cystic fibrosis monitoring. Reproduced with permission[95]. Copyright 2016, Springer Nature Limited. **c** Diaper-integrated urea sensing for jaundice monitoring using a ZnO nanoarray. Reproduced under the terms of the Creative Commons CC BY 4.0 license from ref. 77.

Nonetheless, except for the invasive line, the esophagus is considered the gold standard. Clinicians can also estimate temperature from the environment, where infants housed within an incubator have sensors that measure the ambient temperature. Little research has been conducted to develop a method to non-invasively and accurately measure neonatal core body temperature, but Atallah et al.[66] have succeeded with an unobtrusive method of continuous neonatal brain temperature (a proxy for CBT) monitoring with a zero-heat-flux (ZHF) sensor matrix placed beneath the head[67]. ZHF methodology assumes that if the heat loss from a surface is reduced to zero, the gradient between core and surface temperature will also reduce to zero. By placing thermistors on either side of a thermal resistance material, a control loop can control a heating pad beneath the matrix to heat one thermistor until it matches the skin-interfacing one, thus providing the cerebral temperature[68,69]. Clinically validated in comparison to esophageal measurements, they found moderately high correlations ($r > 0.5$, $p < 0.001$) for most infants, with lower correlations likely due to poor head placement over the sensors. Alternatively, a miniaturized, skin-interfaced temperature sensing system usable by infants using two negative temperature coefficient (NTC) thermistors, one that is insulated within the foam and interfaces with the skin and the other that faces the ambient air[70]. Using single heat flux principles, they were able to estimate core body temperature with a mean difference of −0.05 °C with a 95% LOA of 0.24 °C in comparison to an ingestible temperature sensor. However, their estimation simply adds 1.2 °C to the measured skin temperature and was only tested on three adult subjects, thus further work is needed to observe if this system is usable for infants.

### Electrochemical sensing

Detection of chemical or protein biomarkers from fluidic specimens by patients can provide progressive details for health pre-diagnosis, diagnosis, and prognosis. Typically, neonates admitted to the NICU are subjected to 7.5-17.3 painful procedures a day for laboratory tests[71], with higher risk patients subjected to more frequent blood draws. Neonatal non-invasive thin film biomarker-sensing restricted to sweat, saliva, and urine can replace conventional invasive blood draws with the same sensitivity towards biomarker detection (e.g., electrolytes, metabolites, cell-secreted proteins). The electrochemical sensing principle relies on measuring the charge transfer of captured analyte reactions on a sensing electrode. These charges allow recordable changes in current (voltammetry/amperometry), conductivity (conductometry), and voltage/potential (potentiometry) for use with wearables. For saliva analysis, pacifiers can be an excellent platform for safe saliva sample-collecting (Fig. 5a). Smart pacifiers can be used for detecting glucose[72], sodium, and potassium[73] with amperometric and voltametric enzyme sensing. Glucose tracking is extremely important for neonatal

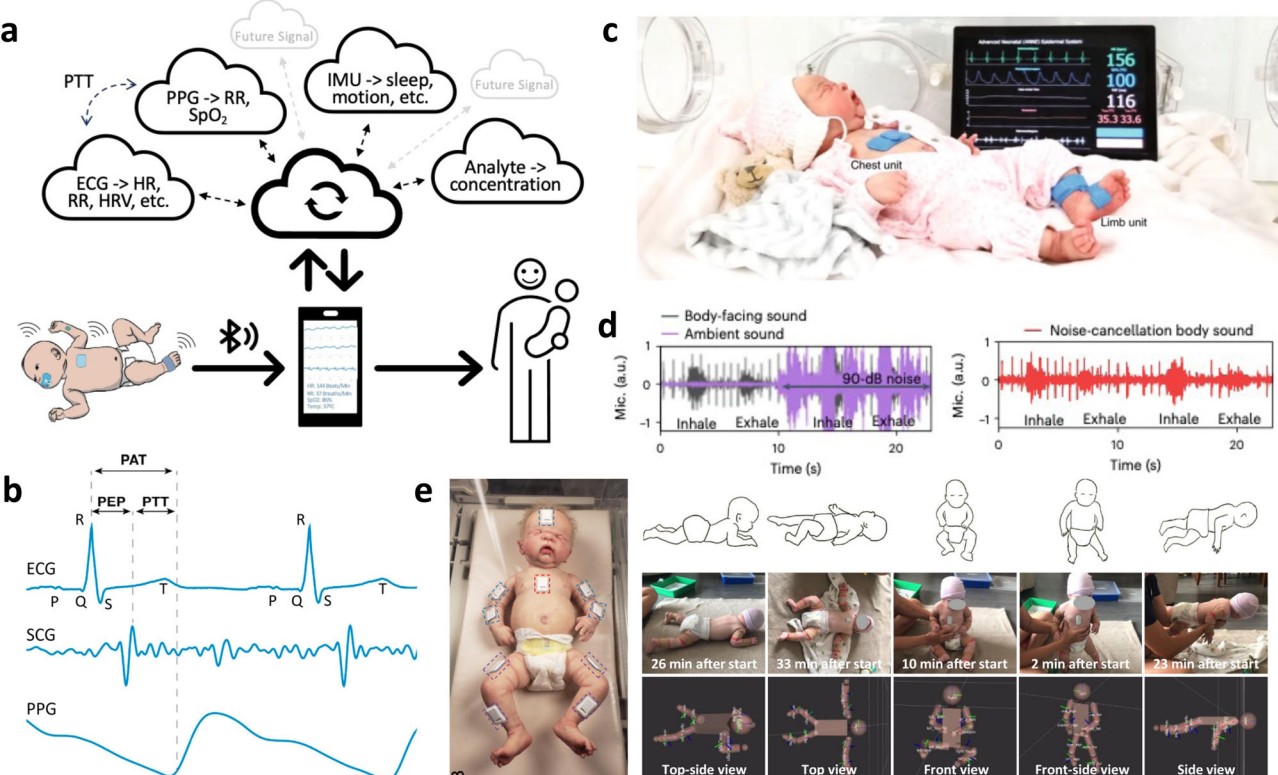

**Fig. 6 | Mechanisms and devices for multi-signal sensing. a** Schematic diagram showing how concurrent and time-synced data uploading and processing allows new physiological phenomena to be monitored. **b** Systolic and diastolic blood pressure derivation from timing differences between fiducials from ECG, SCG, and PPG waveforms. Reproduced under the terms of the Creative Commons CC BY 4.0 license from ref. 96. **c** Binodal multi-signal system that tracks ECG, PPG, SCG, and temperature. Reproduced with permission[17]. Copyright 2020, Springer Nature. **d** Dual-mic setup with one towards the body and another towards the environment allows real-time noise canceling and audio reconstruction (right) even in noisy environment (left). Reproduced with permission[80]. Copyright 2020, Springer Nature. e) Multi-device IMU system (left) that allows body position reconstruction (right). Reproduced with permission[82]. Copyright 2020, Springer Nature.

monitoring, with admitted patients typically subjected to hourly glucose tests and contribute to more than 40% of all laboratory studies performed[71]. Garcia-Carmona, et al.[72] developed a pacifier with a rectifier system to enable forward saliva flow towards an electrochemical cell located in the back without backflow. Their system had excellent linearity ($R^2 = 0.994$), sensitivity ($0.69 \pm 0.04$ nA/mM), limits of detection (0.721 mg/dL), and limits of quantification (1.802 mg/dL) which are far below average glucose levels for neonates (70–150 mg/dL). The current prototype is limited by the biofouling characteristics of saliva affecting the long-term stability of the sensing electrode, but if addressed appropriately, can be effective for continuous glucose monitoring. Parilla et al.[74] designed a saliva-based wearable that aids in the monitoring of phenylalanine, the biomarker for phenylketonuria, a rare inheritable disorder that can be toxic to the nervous system and is tested for at birth. Their sensor has an affordable and user-friendly sampling strategy where saliva is absorbed by a filter paper pre-impregnated with a hydrogen carbonate buffer. The sample then gets placed onto a screen-printed electrode that is linked to a smart wristband that reports the phenylalanine levels after 15–240 s. This sensor has a dynamic range from 0.0004–18 mg/dL which spans below the normal levels (<2 mg/dL) to beyond the unhealthy limits (4 mg/mL) for neonates and can be a favorable alternative to conventional blood draws. For sweat analysis, patch-type sensors can be used to detect and analyze glucose. Open circuit potentiometry can identify the target ion concentration, and the monitoring of sodium and chloride contents trends can support the early detection of cystic fibrosis (Fig. 5b)[75]. Although not a direct indicator of serum glucose levels, sweat glucose can track the tendency patterns, which could be beneficial for neonatal diabetes, where Lee et al.[76] developed a stretchable, ultra-thin, patch to facilitate drug delivery decisions. These devices utilize an enzymatic biosensing method within a chronoamperometric setup. From

urine analysis, diapers are a simple wearable platform. Ning et al.[77] created a urea-based bilirubin detector for neonatal jaundice with hydrovoltaic-biosensing on a ZnO nanoarray (Fig. 5d). It outputs a greater voltage the stronger the bilirubin concentration and proportionally powers a series of LEDs for visualization. The diaper has repeatable performance with multiple exposures spread over several hours and is expected to be discarded with the diaper. We expect much growth in wearable analyte-detection devices for neonatal applications, as there is an abundance of work being done with adult applications. However, there is a lack of defined relationships between these biofluids and serum ion concentrations, thus non-invasive diagnostic powers are limited.

## Multi-signal systems

The synchronous recording of a multi-signal system being uploaded to a single data sink allows comparative relationships to form between the signals, driving new physiological metrics to be computed (Fig. 6a). For example, the aforementioned work by Chung et al.[33] had a binodal ECG and PPG measuring system. Because both signals were recorded and saved concurrently, they were able to identify the time elapsed between the ECG R peak to the PPG valley fiducial to derive the pulse arrival time (PAT) (Fig. 6b). They used the Moens-Korteweg equation to demonstrate a linear relationship between 1/PAT and systolic blood pressure (BP); however, their testing was conducted on one adult and with limited statistical validation compared to a cuff monitor. Improving upon their previous design, a year later, Chung et al.[17] developed a thicker, yet still conformal, binodal system with an added high-sensitivity tri-axial accelerometer for posture recognition and cry analysis (Fig. 6c). From the accelerometer, they were also able to measure seismocardiograms (SCG), which is the recording of the physical vibrations of the heartbeat through the chest wall. SCG provides a

new perspective to cardiovascular monitoring because it can distinguish key cardiological events including valve openings, closings, fillings, and ejections. The time delay between the tallest SCG peak representing aortic opening with the PPG valley provides the pulse transit time (PTT). Combined with a known distance between both sensors and vascular assumptions about arterial wall thickness, modulus, and blood density, a surrogate of systolic and diastolic blood pressure (BP) can be derived. Comparing their system to values from invasive arterial lines of two patients, they were able to generate their own calibration curves for BP with respect to pulse arrival/transit time. Testing with additional subjects showed their system performed within the ANSI/AAMI Sp10 requirements for blood pressure cuffs, which requires a mean difference <5 mmHg and standard deviation <8 mmHg. While the ability to determine BP noninvasively and continuously is incredibly beneficial, it is an extremely volatile metric depending on emotional and active states, body position, temperature, stress, and genetic history, thus requiring frequent calibration with a gold standard (invasive arterial line or oscillometric cuff) to maintain accuracy. Additionally, it is important to be cautious relying on metrics derived from timing differences between signals because data time synchronization can be unreliable. Lagging and imprecise time stamps can lead to poor relational data, which for BP estimation from PTT, even an erroneous hundredth of a second can be detrimental as the average PTT times of a 3-month-old is $140 \pm 11$ ms[78] (assumed even shorter for preterm neonates). Chung, et al.[17] navigated this issue by having their chest unit transmit its 16 MHz local clock information to the limb unit, eliminating drift to enable time syncing of less than 1 ms. The combination of ECG and SCG can also be used to estimate stroke volume of patients with congenital heart defects[79]. Yoo et al.[80] created a multi-signal system that combines an inertial measurement unit (IMU) with two microphones to acousto-mechanically monitor cardiorespiratory and gastrointestinal events. Combining the data recorded from each sensor allows spaciotemporal mapping of the lungs and bowels. The two microphones face opposite directions, one towards the body and other towards the environment, which allow sound separation and audio reconstruction of physiological sounds in noisy environments (Fig. 6d). Simultaneous tracking from multiple of the same sensors can be advantageous as well. A four-IMU system applied to the wrists and ankles combined with machine learning has found high correspondence to cramped-synchronized general movements as compared to visually validated videos for cerebral palsy detection[81]. Additional IMUs can be used to reconstruct and track 3D body motions[82] (Fig .6e).

## Future work

Most of the technology discussed in this review has been developed to simplify and improve treatment options within the NICU. Table 1 shows the comparative wearable methods to conventional physiological monitoring. Notably, there are many adult systems that we refrained from discussing because they had yet to be applied to neonatal and pediatric applications. However, we will share several developments that we find worthwhile but will need to be adapted to accommodate pediatric physiological differences. Wearable stethoscope technology[80,83] could be used to aid in asthma monitoring and continuous studying of cardiopulmonary sounds. Wearable dry cEEG, like the cap device by Mullen et al.[40], could greatly improve ease of seizure monitoring for preterm infants. Wearable piezoelectric ultrasound patches could assist with complex and expensive internal imaging[84]. Use of continuous electrochemical sensing that are currently developed but not in a wearable form factor would be beneficial to help detect and monitor ailments like lactic acidosis[85], neonatal sepsis[86], and jaundice recovery[87]. The application of soft, flexible, and inexpensive passive sensors could help to monitor physiological metrics and movement[88,89]. It is also important to solve known deficiencies, like developing a more accurate pulse oximetry algorithm for hypoxic neonates and infants of various skin tone, as current commercial devices are often inaccurate for patients with blood saturation below 85% or dark skin.

Although the work on pediatric wearables is relatively new, with advanced multimodal systems being developed only within the past five years, it would be extremely valuable to integrate them into real hospital settings for monitoring. Though there is may exist a lack of trust in the efficacy in these devices, hesitating to implement them as tools for influential medical decision making, neonatal and infant patient groups are the most to benefit from this technology. With their fragile skin and risk for severe injury even from acrylic adhesive tapes with weak adhesion forces like paper tape and Tegaderm, the gentle adhesion from elastomers with Van der Waals forces ensure protected skin integrity. Wireless devices reduce the overwhelming visual stress of NICU monitoring systems making parents feel more comfortable to approach and hold their sick child. They also make it easier to transport patients between wards, hospitals, or even from the bed to their parent's arms for kangaroo care[90]. Kangaroo care is critically important for newborn development, parent-baby bonding, and improving patient outcomes, especially for high-risk preterm neonates, with benefits for the infant including more stable physiological metrics, less pain, better sleep, improved weight gain, and earlier discharge[90–92]. Recent wearable technology has also been designed to account for sanitization in the hospital environment, either prepared for autoclave sterilization[17] or to be low cost for economic single-use[93]. The use of machine learning algorithms may aid in the detection of unusual behavior, with highly accurate systems being especially beneficial in remote or lower-resource areas with less-experienced physicians, increasing accessibility to quality healthcare. Combined with cloud-based processing and data storage, long-term patient-specific trend analysis could be manipulated to help clinicians recognize patient recovery/decline, discern abnormal deviations, and reduce misdiagnoses. Concerns for high-fidelity data transfer without dropouts or loss of connection are problems that should be addressed to improve trust in wireless systems, as they are highly likely in hospital environments with lots of devices communicating in the same frequency spectrum as BLE. However, operating with different communication protocols like ultra-wideband transmission may reduce signal interference from other devices while still operating at low power.

The wearable devices developed would provide even greater benefit to outpatient care and use in lower-resource communities. While still mostly in their exploratory and validations phases, the market lacks adequate and trustworthy home infant health monitoring systems. A drive for commercialization allowing normal use will be especially beneficial for recovery monitoring of post-surgical high-risk patients like interstage single-ventricle patients. At the same time, it is important to design user-friendly devices to be able to entrust nonmedically trained individuals to record high-quality data. Battery-free systems with passive sensors often use induction-based data transfer to receiver units. If embedded within clothes, it could standardize and improve ease of use by non-physician caretakers. With the rise of telehealth, low-cost reusable and disposable devices could improve healthcare accessibility, allowing remote or rural patients to conduct a pre-screening at home before deciding to travel long distances to hospitals with specialized care options. It could also reduce strain on hospital resources, allowing low-risk patients to be monitored remotely in the comfort of their own homes. These automated monitoring systems would be especially beneficial for diseased school-age children to improve independence, allowing them to be more self-sufficient and in tune with their medical needs during the long periods of the day they spend with reduced adult supervision. It is important to ensure proper and limited usage of these devices, as they can provide a false sense of safety and increase parental anxiety.

## Conclusion

The development of flexible electronics and wearable health monitoring systems has been a growing field that greatly benefits a vulnerable and understudied population. They can assist in the measurement of key physiological metrics, including cardiovascular operation and rhythm, cerebral hemodynamics, blood oxygen saturation, temperature, and ion concentrations, replacing invasive procedures and improving patient outcomes. Other advanced signal processing allows further medical derivations including blood pressure, core body temperature, and reconstructive imaging of body movements. The information combined from these metrics

can help with diagnostic practices and recovery monitoring; however, the work discussed has only brushed the surface of possible biomedical applications for neonatal populations. With further research and development tuned specifically for infants, this demographic may be the largest beneficiaries from what wearable biosensors and smart health monitoring systems have to offer.

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

## Acknowledgements

The authors acknowledge the support of the National Institutes of Health (Grant No. R21EB034893). This work was also supported by the Imlay Foundation – Innovation Fund.

## Author contributions

Conceptualization, L.Z. and W.-H.Y.; writing—original draft preparation, L.Z., M.G., K.R.K. and W.-H.Y.; writing—review and editing, L.Z., M.G., K.R.K. and W.-H.Y.; supervision, W.-H.Y.; project administration, W.-H.Y.; funding acquisition, W.-H.Y. All authors have read and agreed to the published version of the manuscript.

## Competing interests

The authors declare no competing interests.
