## [Peer Review File · Communications Materials]

19th Feb 24

Dear Professor Yeo,

Thank you for submitting your manuscript, "Wearable, Skin-Interfaced, Non-Invasive Biosensors and Smart Health Monitoring Systems for Infants and Neonates: A Review", to Communications Materials. It has now been seen by 2 referees, whose comments are appended below. You will see that while they find your work of interest, some important points are raised.

In particular, you will see that both referees request additional information on the relevance of the described technologies for infant monitoring. We are not strict on the allowed word limit, and we feel that the requested revisions from the referees would substantially improve the paper.

We are interested in the possibility of publishing your study in Communications Materials, but would like to consider your response to these concerns in the form of a revised manuscript before we make a decision on publication.

We therefore invite you to revise and resubmit your manuscript, taking into account the points raised.

When submitting your revised manuscript, please include the following:

-A response letter with a point-by-point reply to each of the referee comments and a description of changes made. Please include the complete referee report in the response letter. Please note that the response letter must be separate to the cover letter to the editors.

-A marked-up version of the manuscript with all changes to the text in a different colored font. Please do not include tracked changes or comments. Please select the file type 'Revised Manuscript - Marked Up' when uploading the manuscript file to our online system.

-A clean version of the manuscript. Please select the file type 'Article File'.

-An updated Editorial Policy checklist, uploaded as a 'Related Manuscript File' type. This checklist is to ensure your paper complies with all relevant editorial policies. If needed, please revise your manuscript in response to these points. Please note that this form is a dynamic 'smart pdf' and must therefore be downloaded and completed in Adobe Reader. Clicking this link will download a zip file containing the pdf.

In the event that your manuscript is accepted we will provide detailed guidance on our journal policies and formatting. You may however wish to ensure that the manuscript complies with our house style at this stage. See our style and formatting guide (<https://www.nature.com/documents/commsj-phys-style-formatting-guide-accept.pdf>) and checklist (<https://www.nature.com/documents/commsj-phys-style-formatting-checklist-article.pdf>) for reference.

Please use the following link to submit your documents:

[link redacted]

We hope to receive your revised paper within three months; please let us know if you aren't able to submit it within this time so that we can discuss how best to proceed. If we don't hear from you, and the revision process takes significantly longer, we will close your file. In this event, we will still be happy to reconsider your paper at a later date, as long as nothing similar has been accepted for publication at Communications Materials or published elsewhere in the meantime.

Please do not hesitate to contact me if you have any questions or would like to discuss these revisions further. We look forward to seeing the revised manuscript and thank you for the opportunity to review your work.

Best regards,

Rona Chandrawati, PhD
Editorial Board Member
Communications Materials
orcid.org/0000-0002-9780-8844

Reviewers' comments:

Reviewer #1 (Remarks to the Author):

In this article, the authors review wearable sensors designed for health monitoring in infants and neonates. The Introduction section provides a clear overview of the key issues. These issues include usability of the sensors from the perspective of the user (e.g., caregivers, clinicians, infants) and validity of the data obtained from the sensors. Usability involves comfort, ease of use, and ability to perform needed activities, such as picking up the infant (kangaroo care). Validity of the sensor data involves the degree to which the data match what would be obtained when using gold-standard or current standard-of-care equipment. Additional issues briefly mentioned include power consumption (and ability to record continuously) and scalability. Each of the issues are critical to assessing the clinical utility of a given sensor or sensor system. Throughout the review, I was expecting these specific issues to be addressed in the context of use with infants specifically.

The authors address the first issue of usability in the first section on "material developments". The authors consider design considerations for neonates and focus on skin-related risks in current standard of care. These issues are important and were covered especially well in the subsection on design considerations for neonates; it was less clear how the other sections on EES, all in one systems, and textile centers related to infants specifically. Also, throughout the first main section, greater clarity on degree to which these materials have been used in clinical settings and specific

findings related to testing with infant populations would help to strengthen the impact of this paper.

Summarizing specific metrics and providing critical evaluation of the current state of the art in each wearable type would also strengthen section 2 (Physiological Monitoring). The authors should consider providing additional information in Table 1 that not only lists the “wearable alternative” but provides information on how it performs (both in terms of usability and data validity) to the current standard of care or “conventional method.” Also, the first and only reference to Table 1 is at the beginning of the Future Work section. I would recommend referencing it much earlier in the paper.

In the second main section on physiological monitoring, the authors provide general overviews of different monitoring modalities, including ECG, EEG, PPG, NIRS, and temperature. As such, there is a lot of breadth to this article, and I wondered whether the literature reviewed for wearables in each modalities (and as reflected in the references shown in Table 1) was exhaustive. In short, many questions remained about the extent to which the sensors described were developed, validated, and/or used clinically. Only toward the very end of the paper do the authors provide some general answers to these questions, but more pointed and specific critiques of each sensor modality in section 2 would provide the reader with much needed context.

The authors are covering a lot of material in this review. They might consider where they can make their strongest contributions to the literature (also see <https://doi.org/10.1038/s44222-023-00090-0> for recent review on a similar topic) and remove sections that are not directly related to the main issues that are well laid out in the beginning section of the paper. For instance, the authors at times reference machine learning algorithms to automate detection of certain physiological states or abnormalities, but this seems to be a very large and separate topic. The authors might consider focusing solely on the sensor materials and modalities, especially given that such a focus aligns more closely with the mission of this particular journal.

Reviewer #2 (Remarks to the Author):

The review topic is interesting for the recent wearable sensor systems development in the adult patients or elders. However the paper structure is difficult to be understood, the two sections of the materials and physiological monitoring systems should be explained with more information about the relationship to Infants.

Reviewer #1 (Remarks to the Author):

In this article, the authors review wearable sensors designed for health monitoring in infants and neonates. The Introduction section provides a clear overview of the key issues. These issues include usability of the sensors from the perspective of the user (e.g., caregivers, clinicians, infants) and validity of the data obtained from the sensors. Usability involves comfort, ease of use, and ability to perform needed activities, such as picking up the infant (kangaroo care). Validity of the sensor data involves the degree to which the data match what would be obtained when using gold-standard or current standard-of-care equipment. Additional issues briefly mentioned include power consumption (and ability to record continuously) and scalability. Each of the issues are critical to assessing the clinical utility of a given sensor or sensor system. Throughout the review, I was expecting these specific issues to be addressed in the context of use with infants specifically.

Comment #1: The authors address the first issue of usability in the first section on “material developments“. The authors consider design considerations for neonates and focus on skin-related risks in current standard of care. These issues are important and were covered especially well in the subsection on design considerations for neonates; it was less clear how the other sections on EES, all in one systems, and textile centers related to infants specifically. Also, throughout the first main section, greater clarity on degree to which these materials have been used in clinical settings and specific findings related to testing with infant populations would help to strengthen the impact of this paper.

Response: Dear reviewer, thank you for your insightful recommendation. Following the reminder that this work is not under a strict word count, I took care to fill in the gaps you noticed and agree that they have improved this review substantially—thank you! For the sections on EES, all-in-one systems, and textile sensors, I have emphasized how these advancements address the weaknesses I mentioned in the neonatal design considerations. This includes gentle yet accurate monitoring capable from EES and improved maneuverability and infant-caregiver skin contact from all-in-one systems. For the textile sensors portion, I tried to reframe it as being a substantial part of the existing literature/technology for neonatal monitoring, but that we have now progressed beyond those methodologies to move towards soft and flexible electronics.

Manuscript modifications:

- Section 1.2: Emphasized how EES mechanical properties allow comfortable and gentle monitoring on fragile infant skin while maintaining high signal quality.
- Section 1.2: Gave greater background on common tapes used in the NICU and how they are damaging to neonatal skin, providing greater advantages towards EES.
- Section 1.3: Emphasized how neonatal patient care could benefit strongly from all-in-one systems.
- Section 1.4: Gave greater context as to why referencing textile sensors and how the rest of the review will focus on soft and flexible electronics.

Comment #2: Summarizing specific metrics and providing critical evaluation of the current state of the art in each wearable type would also strengthen section 2 (Physiological Monitoring). The authors should consider providing additional information in Table 1 that not only lists the “wearable alternative” but provides information on how it performs (both in terms of usability and data validity) to the current standard of care or “conventional method.” Also, the first and only reference to Table 1 is at the beginning of the Future Work section. I would recommend referencing it much earlier in the paper.

Response: Dear reviewer, thank you for this suggestion. I have now referenced Table 1 in the introduction of the Physiological Monitoring section. I agree that having the table be a referenceable figure early on would be a more helpful tool to guide the reader for the rest of the review. For Table 1, I added a column that compares the performance of the wearable to the conventional method. Most of the wearable alternatives had high correlation to the gold standard, but I noted the works that needed further clinical validation.

Manuscript modifications:

- Section 2: Referenced Table 1 earlier in the article.
- Adapted Table 1:

Metric/Test	Conventional Method	Wearable Alternative	Performance Compared to Conventional	Reference
Heart rate, heart rate variability, heart rhythm	Wired gel electrodes	Wireless ECG	High correlation	[15, 17, 32, 33]
		Wireless pulse oximeter	High correlation, but sensitive to motion artifacts	[47, 48]
Respiration	Wired gel electrodes	Wireless ECG	High correlation	[15, 17, 32, 33]
		Wireless pulse oximeter	High correlation, but sensitive to motion artifacts	[44]
Blood oxygen saturation, hemoglobin	Wired probe to pulse oximeter	Wireless pulse oximeter	High correlation	[15, 33, 47, 48]
Cerebral hemodynamics	Cerebral ultrasound, functional MRI (fMRI), CT scan	Wireless NIR	High correlation with fMRI	[58]
Core Temperature	Invasive arterial line, Rectal/esophageal probe	ZHF sensor	Moderately high correlation to esophageal probe	[66]
		Transistor	Needs further validation	[70]
Blood Pressure	Invasive arterial line	Wireless Multi-signal system	High correlation but frequent calibration is required	[15, 33]
Cerebral Palsy Detection	MRI, Wired EEG	Wireless Multi-signal system	Moderate accuracy, further clinical validation is necessary	[58, 81, 82]

Jaundice detection	Blood draw/urine test	Wireless optical colorimetry	High correlation	[59]
		Wireless noninvasive electrochemical diaper (urine)	Not clinically validated	[77]
Cystic fibrosis detection (Chloride monitoring)	Blood draw	Wireless noninvasive electrochemical sensor patch (sweat)	High correlation	[75, 93]
Phenylketonuria Detection	Blood draw	Wireless noninvasive electrochemical wristband (saliva)	High correlation, further clinical validation is necessary	[74]
Glucose monitoring	Blood draw/urine test	Wireless noninvasive electrochemical sensor patch (sweat)	High correlation	[75, 76]
Saliva sampling	Swab	Wireless noninvasive electrochemical pacifier (saliva)	High correlation	[72]
Sodium monitoring	Blood draw	Wireless noninvasive electrochemical pacifier (saliva)	High correlation	[73]
Potassium monitoring	Blood draw	Wireless noninvasive electrochemical pacifier (saliva)	High correlation	[73]

Comment #3: In the second main section on physiological monitoring, the authors provide general overviews of different monitoring modalities, including ECG, EEG, PPG, NIRS, and temperature. As such, there is a lot of breadth to this article, and I wondered whether the literature reviewed for wearables in each modality (and as reflected in the references shown in Table 1) was exhaustive. In short, many questions remained about the extent to which the sensors described were developed, validated, and/or used clinically. Only toward the very end of the paper do the authors provide some general answers to these questions, but more pointed and specific critiques of each sensor modality in section 2 would provide the reader with much needed context.

Response: Dear reviewer, thank you for this recommendation. I have gone back and expanded on the developments that I was too succinct on previously and have provided more pointed weaknesses in hopes that they help drive future work. I have made sure to include the clinical validation equivalent for applications that have one. Some NIRS, electrochemical, and multi-signal applications lack clinical gold standards.

Manuscript modifications:

- Section 2.1.2: Provided greater context on the benefit of a hybrid electrode design.
- Section 2.1.2: Noted weakness of bristle electrode.

- Section 2.3: Noted clinical validation method for the ZHF temperature sensor for core body temperature estimation.
- Section 2.3: Noted clinical validation and weakness of thermistor sensor for core body temperature estimation.
- Section 2.4: Gave greater context to the benefit of noninvasive electrochemical sensing for neonates, as it will reduce large the amount of blood draws common for patients in the NICU.
- Section 2.4: Provided sensor characterization for the glucose-sensing smart pacifier.
- Section 2.4: Provided sensor characterization for the phenylalanine-sensing smart wearable wristband.
- Section 2.4: Provided more information on the operation of the bilirubin-sensing diaper.
- Section 2.5: Provided clinical validation methods and weaknesses for real-time blood pressure determination.

Comment #4: The authors are covering a lot of material in this review. They might consider where they can make their strongest contributions to the literature (also see <https://doi.org/10.1038/s44222-023-00090-0> for recent review on a similar topic) and remove sections that are not directly related to the main issues that are well laid out in the beginning section of the paper. For instance, the authors at times reference machine learning algorithms to automate detection of certain physiological states or abnormalities, but this seems to be a very large and separate topic. The authors might consider focusing solely on the sensor materials and modalities, especially given that such a focus aligns more closely with the mission of this particular journal.

Response: Dear reviewer, thank you for your suggestions. Though I do cover several breadth of topics in this review, I want to assure that I tried to reference as much of the existing research that has been conducted specifically for neonatal patient groups (which is limited). While there exist many developments for adults that could be applicable for infants, I primarily shared work that specifically targeted/included neonatal patients and focused on the most novel and recent advancements. I reduced sections about machine learning from the physiological measurements section as they do deter from the mission of this journal, but I have decided to keep my recommendations in the future works section as wireless communication, cloud-based data storage, and machine learning are driving what sensor modalities are worth exploring. Again, thank you so much for your comments! I think that they really helped strengthen the possible impact of this review.

Manuscript modifications:

- Section 2.1.1: Removed unnecessary discussion on machine learning in this section.

Reviewer #2 (Remarks to the Author):

The review topic is interesting for the recent wearable sensor systems development in the adult patients or elders. However the paper structure is difficult to be understood, the two sections of the materials and physiological monitoring systems should be explained with more information about the relationship to Infants.

Response: Dear reviewer, thank you for this recommendation. I tried to explain in greater depth the purpose of the review and its structure. Namely, in the introduction of each section I try to be more clear about the patient group that paper is focused on and provide a summary of what to expect.

Manuscript modifications:

- Introduction: More explicit about article structure and its relationship to infants.

10th Mar 24

Dear Professor Yeo,

Your manuscript titled "Wearable, Skin-Interfaced, Non-Invasive Biosensors and Smart Health Monitoring Systems for Infants and Neonates: A Review" has now been seen again by our referees, whose comments appear below. In light of their advice I am delighted to say that we are happy, in principle, to publish a suitably revised version in Communications Materials under the open access CC BY license (Creative Commons Attribution v4.0 International License).

We therefore invite you to edit your manuscript to comply with our journal policies and formatting style in order to maximise the accessibility and therefore the impact of your work.

EDITORIAL REQUESTS

* Your manuscript should comply with our policies and format requirements, detailed in our style and formatting guide (<https://www.nature.com/documents/commsj-phys-style-formatting-guide-accept.pdf>).

* Please edit your manuscript according to the editorial requests in the attached table, and outline revisions made in the right hand column. If you have any questions or concerns about any of our requests, please do not hesitate to contact me. It is important that each request be addressed in order to avoid delays in accepting your manuscript. Please upload the completed table with your manuscript files as a Related Manuscript file.

* The editorial requests table also includes a full list of the files that must be provided upon resubmission. Please upload your files according to this table.

* An updated editorial policy checklist that verifies compliance with all required editorial policies must be completed and uploaded with the revised manuscript. All points on the policy checklist must be addressed; if needed, please revise your manuscript in response to these points. Please note that this form is a dynamic 'smart pdf' and must therefore be downloaded and completed in Adobe Reader. Clicking this link will download a zip file containing the pdf.

OPEN ACCESS

Communications Materials is a fully open access journal. Articles are made freely accessible on publication under a CC BY license (Creative Commons Attribution 4.0 International License). This license allows maximum dissemination and re-use of open access materials and is preferred by many research funding bodies.

RESUBMISSION

At acceptance, you will be provided with instructions for completing this CC BY license on behalf of all authors. This grants us the necessary permissions to publish your paper. Additionally, you will be asked to declare that all required third party permissions have been obtained.

Please use the following link to submit your revised files:

[link redacted]

We hope to hear from you within two weeks; please let us know if the process may take longer.

Best regards,

Rona Chandrawati, PhD
Editorial Board Member
Communications Materials
orcid.org/0000-0002-9780-8844

REVIEWERS' COMMENTS:

Reviewer #1 (Remarks to the Author):

I appreciate the authors' clear and specific responses to my prior comments. This paper makes a strong contribution in reviewing state of the art in wearables for infants and neonates. I have no additional comments.

Reviewer #2 (Remarks to the Author):

Authors well addressed the questions from reviewers, and the paper is acceptable for the journal publication.